# The bHLH-zip transcription factor SREBP regulates triterpenoid and lipid metabolisms in the medicinal fungus *Ganoderma lingzhi*

Yong-Nan Liu [iD] [1,2,3], Feng-Yuan Wu[1,2,3], Ren-Yuan Tian[1,2,3], Yi-Xin Shi[1,2,3], Zi-Qi Xu[1,2,3], Ji-Ye Liu[1,2,3], Jia Huang[1,2,3], Fei-Fei Xue[1,2,3], Bi-Yang Liu[1,2,3] & Gao-Qiang Liu [iD] [1,2,3✉]

Ganoderic acids (GAs) are well recognized as important pharmacological components of the medicinal species belonging to the basidiomycete genus *Ganoderma*. However, transcription factors directly regulating the expression of GA biosynthesis genes remain poorly understood. Here, the genome of *Ganoderma lingzhi* is *de novo* sequenced. Using DNA affinity purification sequencing, we identify putative targets of the transcription factor sterol regulatory element-binding protein (SREBP), including the genes of triterpenoid synthesis and lipid metabolism. Interactions between SREBP and the targets are verified by electrophoretic mobility gel shift assay. RNA-seq shows that SREBP targets, mevalonate kinase and 3-hydroxy-3-methylglutaryl coenzyme A synthetase in mevalonate pathway, sterol isomerase and lanosterol 14-demethylase in ergosterol biosynthesis, are significantly upregulated in the SREBP overexpression (OE::SREBP) strain. In addition, 3 targets involved in glycerophospholipid/glycerolipid metabolism are upregulated. Then, the contents of mevalonic acid, lanosterol, ergosterol and 13 different GAs as well as a variety of lipids are significantly increased in this strain. Furthermore, the effects of SREBP overexpression on triterpenoid and lipid metabolisms are recovered when OE::SREBP strain are treated with exogenous fatostatin, a specific inhibitor of SREBP. Taken together, our genome-wide study clarify the role of SREBP in triterpenoid and lipid metabolisms of *G. lingzhi*.

[1] Hunan Provincial Key Laboratory of Forestry Biotechnology, Central South University of Forestry & Technology, Changsha, Hunan 410004, China. [2] International Cooperation Base of Science and Technology Innovation on Forest Resource Biotechnology of Hunan Province, Central South University of Forestry & Technology, Changsha, Hunan 410004, China. [3] Microbial Variety Creation Center, Yuelushan Laboratory of Seed Industry, Changsha 410004, China. ✉email: gaoliuedu@csuft.edu.cn

Fungi are extraordinary organisms that readily create a diverse set of natural products called secondary metabolites, some of which are beneficial to humans[1]. *Ganoderma* (Ganodermaceae) is a genus of edible and medicinal mushroom that is known as "Reishi" in Japanese and "Lingzhi" in Chinese. The medicinal values of this mushroom were documented thousands of years ago[2]. Modern chemistry and pharmacology has demonstrated that ganoderic acids (GAs), which are a group of highly oxygenated lanostane-type triterpenoids, is the key pharmacologically active compounds of *Ganoderma* species[3]. A lot of studies have demonstrated that GAs have various biological functions, such as cytotoxicity to several cancer cells in vitro[4], the inhibition of tumour invasion in vitro and in vivo[5], regulation of osteoclastogenesis[6], protection of the liver[7], and anti-human immunodeficiency virus[8]. Progress has been achieved in enhancing GA production in *Ganoderma* species by the manipulation of fermentation strategies[9,10], addition of chemical inducers[11,12] and genetic engineering[13,14].

In addition to studies on pharmacological analysis and enhancing the production of GA, some work has been conducted to study the regulatory mechanism of GA biosynthesis. To date, the roles of the upstream signalling molecules reactive oxygen species[15,16], $Ca^{2+}$[17], cAMP[18] and membrane phospholipids[19–21] in GA biosynthesis have been preliminarily elucidated. However, the downstream pathways of these known signalling molecules, especially transcription factors that directly regulate the expression of GA biosynthesis genes, remain poorly understood for the regulatory mechanism of GA biosynthesis.

High quality and complete genome annotation information is needed for studying transcription factors binding to promoters and regulating target gene expression. To date, four whole-genome sequencing projects for *Ganoderma lucidum* have been published and registered in the BioProject section of the National Center of Biotechnology Information (NCBI), which contains available coding gene sequences and genome sequence information but lacks available genome annotation information[22–25]. Recently, the *Ganoderma lingzhi* species was identified as a new species distributed in East Asia, and it has been incorrectly considered to be another species, *G. lucidum*, which is distributed in Europe, for many years by morphological studies and analysis of rDNA internally transcribed spacer sequences and additional gene fragments[26]. However, no report about the assembly of a high-quality genome of *G. lingzhi* is available.

In this work, the genome of *G. lingzhi* was de novo sequenced and assembled with the combination of Illumina and PacBio sequencing strategies. We then performed gene function analysis and annotated genes involved in triterpenoid and ergosterol biosynthesis. We further identified a basic helix-loop-helix leucine zipper (bHLH-zip) transcription factor, sterol regulatory element-binding protein (SREBP), by genome-wide DNA affinity purification sequencing (DAP-seq). We identified GA and lipid metabolism genes as the target genes of SREBP. A gel-shift assay was used to analyse SREBP protein interactions with promoter DNA of SREBP targets. Furthermore, we found a significant enhancement of GA and lipid metabolism by gene transcription in the SREBP overexpression strain. Taken together, this study develops foundational genomic and genetic resources that can be used in further molecular genetic analyses and breeding in *G. lingzhi* and clarified a genome-wide bHLH-zip transcription factor—SREBP— that directly regulates triterpenoid and lipid metabolism.

## Results

### Genome sequence analyses of *G. lingzhi*.
We sequenced the genome of the *G. lingzhi* strain using a whole-genome shotgun sequencing strategy. As shown in Table 1, the sequences were

**Table 1 General characteristics of the *G. lingzhi* genome.**

| | |
|---|---|
| Total length (bp) | 49,151,792 |
| Scaffold number | 30 |
| GC content (%) | 55.87 |
| N50 (bp) | 2,338,391 |
| Scaffold Min length (bp) | 36,289 |
| Scaffold Max length (bp) | 5,053,266 |
| Scaffold Average length (bp) | 1,638,393.07 |
| Scaffold Median length (bp) | 1,239,252.00 |
| Total protein-coding genes number | 13,125 |
| Average mRNA length (bp) | 1,955.66 |
| Average CDS length (bp) | 1,451.83 |
| Average of exon number | 6.04 |
| Average of exon length (bp) | 240.48 |
| Average of intron length (bp) | 99.97 |
| Total exon number | 79,238 |
| Total intron number | 66,113 |
| Total intron length (bp) | 6,609,295 |

assembled into 30 scaffolds with a total length of 49.15 Mb. The lengths of scaffolds ranged from 36,289 bp to 5,053,266 bp with an N50 scaffold size of 2.33 Mb, and the overall GC content is ~55.87%. By using K-mer analysis, these scaffolds covered 93.71% of the expected whole genome size (52.45 Mb) with a 0.023% read error rate, indicating the high quality of the genome sequence assembly (Supplementary Fig. 1). In total, 13,125 gene models were predicted, with an average mRNA and CDS length of 1,955.66 and 1,451.83 bp, respectively. On average, each predicted gene contained 6.04 exons, and in total, 79,238 exons were contained in all genes (Table 1).

Nine public databases were used to annotate the function of predicted genes by the NCBI non-redundant, Pfam, NCBI clusters of orthologous groups of proteins, UniProt, Kyoto Encyclopedia of Genes and Genomes, Gene Ontology, Pathway, RefSeq, and InterProScan public databases. Overall, 12,802 genes were annotated to at least one function, accounting for 97.54% of all genes (Supplementary Table 1). The annotation results of nine databases were combined to facilitate subsequent research on gene search and exploration (details are shown in Supplementary Data 1).

### The pathway of triterpenoid and ergosterol biosynthesis.
Triterpenoids are one of the most important secondary metabolites with pharmacological activity. Ganoderic acid (GA), a type of triterpenoid, is an important medicinal component found in *Ganoderma* spp. To speculate on the GA biosynthetic pathway, the genes distributed in the mevalonate pathway (MVA) of the "terpenoid backbone biosynthesis (map00900)" pathway were evaluated, and 11 enzymes encoded by 13 genes existed in *G. lingzhi*, including two squalene monooxygenase genes and two farnesyl diphosphate synthase genes (Supplementary Table 2), which is not exactly consistent with previous reports of *G. lucidum* where the acetyl coenzyme A (CoA) acetyltransferase gene and farnesyl diphosphate synthase gene are each encoded by two genes[22]. We further summarized the potential triterpenoid biosynthesis pathway in *G. lingzhi*, as shown in Fig. 1. Compared with the well-studied upstream catalytic synthesis of lanosterol (MVA pathway), the steps following cyclization are largely unknown but most likely include a series of oxidation, reduction, and acylation reactions by the cytochrome P450 (CYP) superfamily. A total of 147 genes were annotated as CYPs by gene searching, as shown in Supplementary Data 2.

Lanosterol is also the common cyclic intermediate of ergosterol, which is one of the important components of the fungal cell membrane, and the various metabolic pathways of lanosterol are divergent[27]. We screened the *G. lingzhi* genes in the

**Fig. 1 Putative triterpenoid and ergosterol biosynthetic pathway in *G. lingzhi*.** The representative triterpenoids of *Ganoderma* spp were marked in red, such as ganoderic acid A, D, and F. The dotted line with three arrows in ergosterol and GA biosynthetic pathway indicates the multiple enzymatic and speculative steps, respectively. Abbreviated word information is shown in Supplementary Table 2.

ergosterol biosynthesis pathway and found a complete pathway of ergosterol biosynthesis from lanosterol, which included 10 enzymes encoded by 17 genes (Supplementary Table 2). Enzyme-catalysed lanosterol to form zymosterol requires five enzymes (lanosterol 14-demethylase, delta14-sterol reductase, methyl sterol monooxygenase, sterol-4alpha-carboxylate 3-dehydrogenase, keto steroid reductase) to undergo eight steps. Then, zymosterol is further catalysed by five enzymes, sterol 24-C-methyltransferase, sterol isomerase, delta7-sterol 5-desaturase, sterol 22-desaturase, and delta 24-sterol reductase, to form ergosterol (Fig. 1, marked in blue; more detailed information can be found in Supplementary Table 2 and ko00100: steroid biosynthesis). In addition, we also analysed some other steroid biosynthetic enzymes starting with farnesyl diphosphate as a substrate, such as polycis-polyprenyl diphosphate synthase, farnesyltransferase type-1 subunit alpha and beta, endopeptidase, and farnesylcysteine lyase, in the *G. lingzhi* genome (Fig. 1, Supplementary Table 2).

**Identification of a potential bHLH-zip transcription factor that regulates triterpene synthesis.** At present, many upstream signalling molecules and functional proteins have been found to regulate GA biosynthesis in *Ganoderma* spp. However, the transcription factors that directly regulate the expression of triterpene biosynthesis genes are still unknown. The basic helix-loop-helix leucine zipper (bHLH-zip) transcription factor sterol

regulatory element-binding protein (SREBP) is conserved in mammalian, worms, flies, and yeast, and functions in the regulation of sterol homoeostasis and lipid metabolism[28]. Over-expression studies indicate that SREBP positively regulates the expression of many sterol synthesis genes in the MVA pathway, such as 3-hydroxy-3-methylglutaryl CoA reductase (the rate-limiting enzyme of sterol biosynthesis), mevalonate kinase, squalene synthase, and fatty acid synthesis genes, such as fatty acid synthase and long-chain fatty acyl elongase, in mammals[29,30]. Because both sterol and triterpenoid synthesis occurs through the MVA pathway (Fig. 1), we speculated that SREBP may be a transcription factor directly regulating triterpene synthesis in *G. lingzhi*.

We used the SREBP of *Trametes pubescens* (GenBank accession number: OJT12715.1) to perform a homology BLAST against the *G. lingzhi* genome database. One gene (g2373) was identified as being a potential ortholog to SREBP. The SREBP gene of *G. lingzhi* had an ORF of 2532 bp located on scaffold 3 at positions 379,572–382,613 (Supplementary Fig. 2a). The deduced protein contains a bHLH DNA binding motif in its N-terminal domain (residues 197–294 aa, Supplementary Fig. 2b). We conducted a conservative analysis of the bHLH domain based on multiple sequence alignment and showed that bHLHs of *G. lingzhi* were highly conserved in comparison with other known bHLHs in several fungi, *Trametes pubescens*, *Neurospora crassa* and *Stemphylium lycopersici*, and less conserved in comparison with

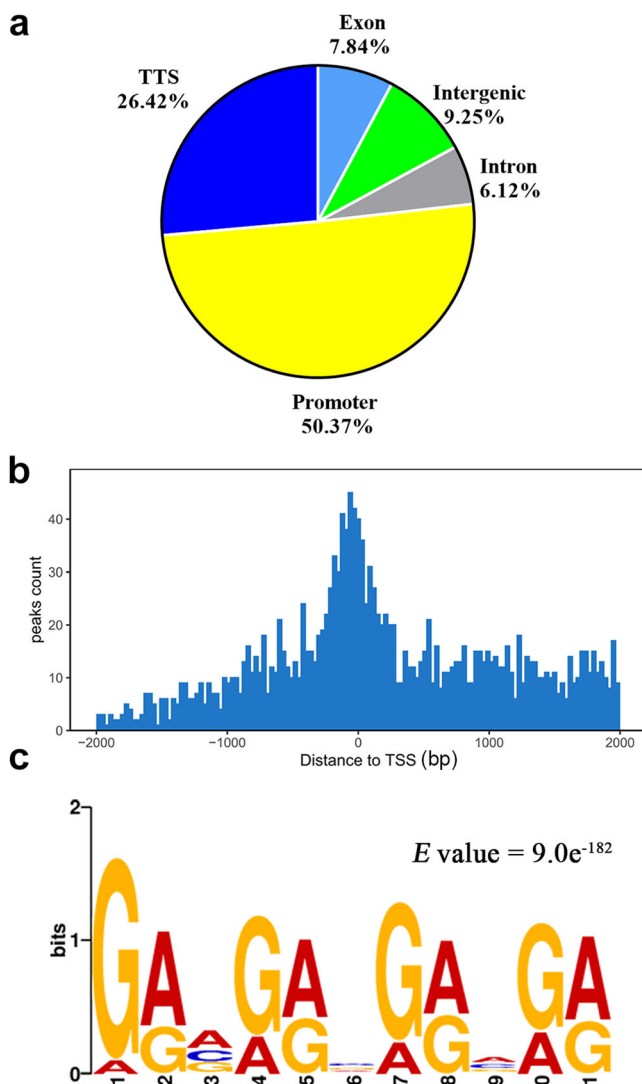

**Fig. 2 Genome-wide identification of SREBP binding sites and motifs.**
**a** Relative binding-peak distribution across genomic regions. **b** Distance to
the transcription start site (TSS) from SREBP binding sites. **c** The 2271 peak
regions were analyzed for overrepresented motifs using MEME. The top-
scoring motif is shown. The other preferred motifs are shown in
Supplementary Fig. 3.

*Mus musculus*, *Homo sapiens*, and *Drosophila willistoni* (Supple-
mentary Fig. 2b). The seven sequences contain 18 conserved
residues, especially tyrosine residue (indicated in red) specific to
the SREBP family of bHLH transcription factors[31]. In summary,
the predicted SREBP protein from *G. lingzhi* has the characteristic
features of SREBP proteins described in other organisms.

**Utilizing DAP-Seq to identify direct target genes of SREBP in
*G. lingzhi*.** To elucidate the potential regulatory mechanism of
SREBP on triterpenoid synthesis in *G. lingzhi*, we used DNA
affinity purification sequencing (DAP-seq) for genome-wide
identification of SREBP binding sites[32]. We identified a total of
2271 putative SREBP binding sites (details are shown in Sup-
plementary Data 3). The binding sites were distributed mostly
within promoter and transcription terminate site (TTS) regions,
accounting for 50.37% (1144) and 26.42% (600), respectively,
with very few sites located in intergenic regions, exons, and
introns, accounting for 9.25%, 7.84%, and 6.12%, respectively
(Fig. 2a). A stretch 2 kb up- and 500 bp downstream of the

transcription start site was defined as promoter[33]. A total of
75.9% of putative SREBP binding sites were located within 2 kb of
a known transcription start site (Fig. 2b). The sequences around
the peaks were then analysed by MEME to search for enriched
sequence motifs[34]. The best-fit core motif was 5′-
GRVGRVGRVGR-3′ ($E = 9.0 \times 10^{-182}$), which was present in
2694 peaks with a *p*-value < 0.0001 (Fig. 2c), followed by 5′-
GCAGAA-3′, 5′-GGCDAC-3′, 5′-GAGATGGGAGAR-3′, and 5′-
AAAASAARAMAA-3′, which were the DNA motifs preferred by
the *G. lingzhi* SREBPs (Supplementary Fig. 3, details are shown in
Supplementary Data 4).

We further searched the 1144 potential promoter binding sites
and identified 1077 potential SREBP target genes of *G. lingzhi*
(additional information on the peak locations and the nearest
gene list is provided in Supplementary Data 3). These target genes
were analysed by KEGG, and clusters for sesquiterpenoid and
triterpenoid biosynthesis (*p* = 0.012), terpenoid backbone bio-
synthesis (*p* = 0.19), glycerolipid metabolism (*p* = 0.58) and
glycerophospholipid metabolism (*p* = 0.40) were enriched as
expected (Fig. 3a; details are shown in Supplementary Data 5).
Several genes of those pathways were found in a KEGG
enrichment analyses pursued a deeper analyses given the interest.
Specifically, three genes, *g4989*, *g3847*, and *g7601*, encode the
same germacrene-A synthase involved in sesquiterpenoid and
triterpenoid biosynthesis. Two genes, *g3941* and *g1941*, encoding
mevalonate kinase and 3-hydroxy-3-methylglutaryl CoA synthe-
tase, respectively, are involved in terpenoid backbone biosynth-
esis. Six genes, *g4280*, *g4309*, *g1208*, *g5041*, *g6347*, and *g805*,
encoding cardiolipin-specific phospholipase, lysophospholipid
acyltransferase (phosphatidylcholine, phosphatidylethanolamine,
and phosphatidic acid biosynthesis), ethanolamine-phospho-
transferase, diacylglycerol kinase, lysophospholipase I, and
triacylglycerol/diacylglycerol lipase, respectively, are involved in
glycerophospholipid/glycerolipid metabolism (Fig. 3a and b). On
the other hand, in the identified SREBP target genes, 10 target
genes belonged to CYPs (*g5000*, *g4875*, *g4164*, *g11088*, *g10251*,
*g3908*, *g11110*, *g6752*, *g87*, and *g7787*), where *g3908*, *g10251* and
*g7787* were involved in ergosterol biosynthesis (Supplementary
Table 3).

Further analysis of the SREBP-matched DNA sequence of the
above target genes involved in terpenoid biosynthesis and lipid
metabolism showed that only one and three SREBPs matched the
DNA sequences of target genes *g7601* and *g6347/1208/g4309*
fitted to the motifs 5′-GGCDAC-3′ and 5′-GAGATGGGAGAR-
3′, respectively. The matched DNA sequences of the other target
genes fitted to the motif 5′-GRVGRVGRVGR-3′. In addition,
most SREBP-matched DNA sequences were located before the
transcription start site, except for *g5041*, *g4039* and *g805* (Fig.
3b). Similar SREBP-matched DNA sequence characteristics also
appeared in the 10 target genes belonging to the CYPs
(Supplementary Table 3).

In order to verify the identified SREBP sites, we further purified
the putative DNA binding domains (bHLHs) of the *G. lingzhi*
SREBP proteins and DNA fragments corresponding to the
promoter regions of the 10 target genes shown in Fig. 3b to
carry out electrophoretic mobility gel shift assays (EMSAs). The
purified bHLH protein could shift all 10 DNA fragments (Fig. 3c,
line 2). The specificity of the SREBP–DNA complex was
confirmed by adding an excess of unlabelled DNA. As expected,
a weakened shift was detected when adding the unlabelled DNA
(Fig. 3c, line 3).

**SREBP overexpression increased triterpenoid, ergosterol and
lipid biosynthesis.** To further clarify the role of SREBP in tri-
terpenoid and lipid metabolism, we characterized the metabolic

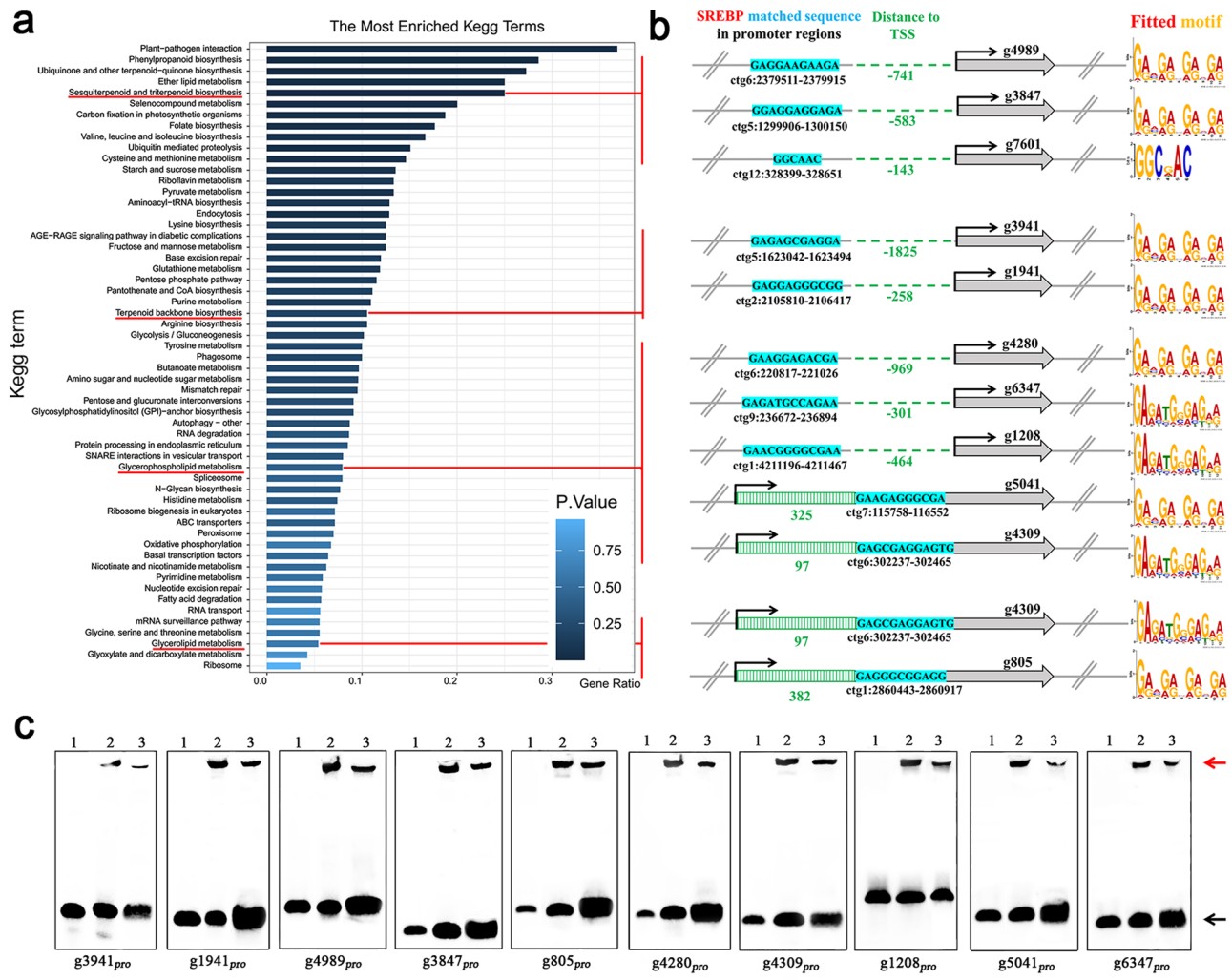

**Fig. 3 Screening of SREBP target genes for terpenoid and lipid metabolism. a** KEGG analysis of SREBP target genes. Clusters for terpenoid and lipid metabolism were highlighted by the red line. **b** Schematic diagram showing the genomic locations of SREBP target genes that involved in terpenoid and lipid metabolism, and corresponding matched DNA sequence, promoter regions, and fitted motif. The conserved SREBP binding sequence was highlighted in blue shadow. **c** SREBP can bind to the DNA sequences in the targets promoter by EMSA. Line 1: the labelled DNA probe control. The labelled DNA probe was preincubated with 0.6 µg SREBP-bHLH protein (Line 2), and an unlabelled competitor DNA probe (200× labelled DNA probe) was added (Line 3). Arrows indicate positions of protein-bound (red) and free (black) DNA bands.

changes of triterpenoids and lipids following overexpression of SREBP by RNA-seq, secondary metabolomics and lipid metabolomics analysis. We constructed a OE::SREBP vector named GLgpd-SREBP that carries the hygromycin B (Hyg) resistance gene as a selectable marker (Supplementary Fig. 4a–c). The transcription levels of SREBP were significantly increased in the transformants determined by RT-qPCR (Supplementary Fig. 4d). A Western blot analysis also showed a significant increase in the protein levels in the OE::SREBP strains. In particular, the nuclear version of SREBP in the OE::SREBP strains was significantly increased ~1.63–2.11-fold of the levels found in the WT strain ($p < 0.01$, Supplementary Fig. 4e, f).

Six RNA libraries from WT and OE::SREBP strains of *G. lingzhi* were sequenced using Illumina paired-end sequencing technology. The average number of clean reads for each sample was 23.20 million (Supplementary Fig. 5a). The mapping ratio of each sample against the reference genome ranged from 91.59% to 92.84% (Supplementary Fig. 5a, b). The mapped reads were assembled and compared with original annotations of the genome. The transcript regions without annotation obtained by the above processes are defined as novel transcripts. Excluding

short transcripts (coding peptides with <50 amino acids) or those containing only one exon, 3996 novel transcripts were discovered in this project (detailed information of the novel transcripts shown in Supplementary Data 6, and has been deposited at NCBI: PRJNA738334, SRR17081370).

A systematic analysis of expression profiles through RNA-seq identified 4601 differentially expressed genes (DEGs, fold change ≥ 2 and false discovery rate < 0.01), including 2152 upregulated and 2449 downregulated genes, in the OE::SREBP strain compared to the WT strain (Fig. 4a, Supplementary Data 7). Venn analysis showed that 177 upregulated DEGs belonged to SREBP target genes by integrating RNA-seq and DAP-seq data (Fig. 4b). The 2152 upregulated DEGs were enriched for functional categories involved in metabolic activities by KEGG. In particular, steroid biosynthesis ($p$-value = $6.18E^{-5}$), sesquiterpenoid and triterpenoid biosynthesis ($p$-value = $8.70E^{-3}$), glycerophospholipid metabolism ($p$-value = $7.35E^{-1}$), glycerolipid metabolism ($p$-value = $8.04E^{-1}$), fatty acid biosynthesis ($p$-value = $2.78E^{-1}$), and sphingolipid metabolism ($p$-value = $4.53E^{-1}$) were enriched as uppathway in OE::SREBP strain compared to WT strain (Fig. 4c and Supplementary

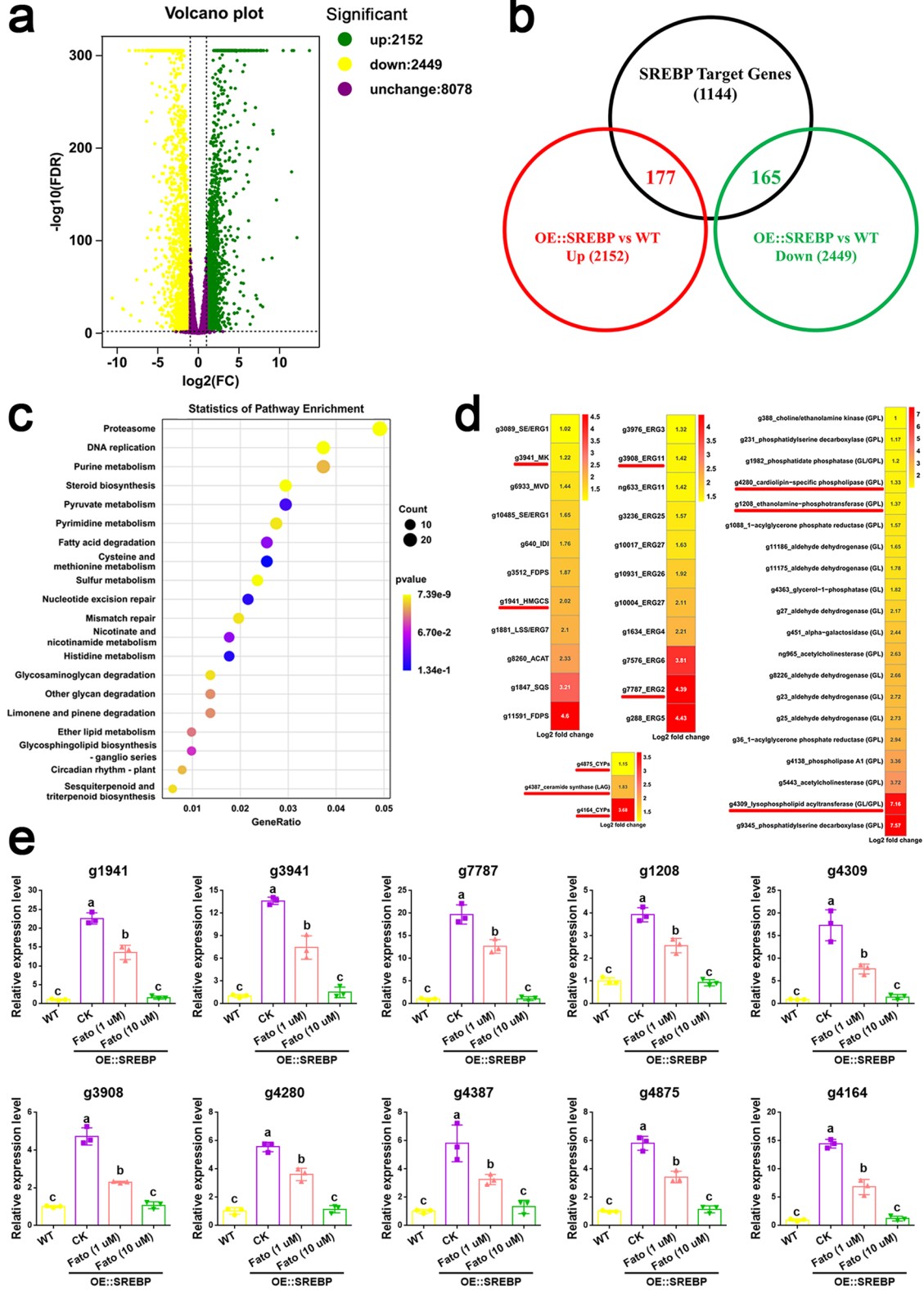

Data 8). Specifically, 22 upregulated DEGs were involved in steroid and triterpenoid biosynthesis, in which 11 DEGs involved in the MVA pathway, such as squalene synthase (*g1847*), lanosterol synthase (*g1881*), and SREBP targets 3-hydroxy-3-methylglutaryl CoA synthetase (*g1941*) and mevalonate kinase (*g3941*) and the other eleven ERGs, such as delta 24-sterol reductase (ERG4, *g1634*) and sterol 22-desaturase (ERG5, *g288*),

and SREBP targets sterol isomerase (ERG2, *g7787*) and lanosterol 14-demethylase (ERG11, *g3908*). In addition, twenty-two upregulated DEGs were involved in glycerophospholipid/glycerolipid metabolism and contained three SREBP targets: ethanolamine-phosphotransferase (*g1208*), lysophospholipid acyltransferase (*g4309*), and cardiolipin-specific phospholipase (*g4280*). We also found that the SREBP targets ceramide synthase (*g4387*) and

**Fig. 4 Transcriptional profiling revealed that the transcription factor SREBP regulates triterpenoid, ergosterol and lipid biosynthesis genes expression in _G. lingzhi_. a** Volcano plot on differential expression genes between WT and OE::SREBP strains. Significant DEGs were defined with the adjusted fold change (FC) ≥ 2 and false discovery rate (FDR) < 0.01. Green-, red- and black-dots are down-regulated, up- regulated and without significant difference, respectively. **b** Venn analysis the SREBP target gene expression levels by integrating RNA-seq and DAP-seq data. **c** KEGG analysis of the up-regulation DEGs in OE::SREBP strain. **d** Differential expression of triterpenoid, sterol and lipid biosynthesis genes ranked by degree of log2 fold change from **a** and **b**. The SREBP direct target genes are shown in red line. The abbreviation information is shown in Supplementary Table 2. **e** qRT-PCR analyses of ten selected SREBP target genes in WT and OE::SREBP strains. 1 μM and 10 μM fatostatin (abbreviated as Fato, CAS 298197-04-3, Simga) were added at shaking for 5 days and then maintained until the 7th days in liquid cultures of OE::SREBP strains. CK indicated the OE::SREBP strains not treated with fatostatin. The expression level of each gene from the WT strains was arbitrarily designated a value of 1. The values are the means ± SD of three independent experiments. Different letters indicate significant differences between the lines (n = 3 independent experiments, P < 0.05, according to Duncan's multiple range test).

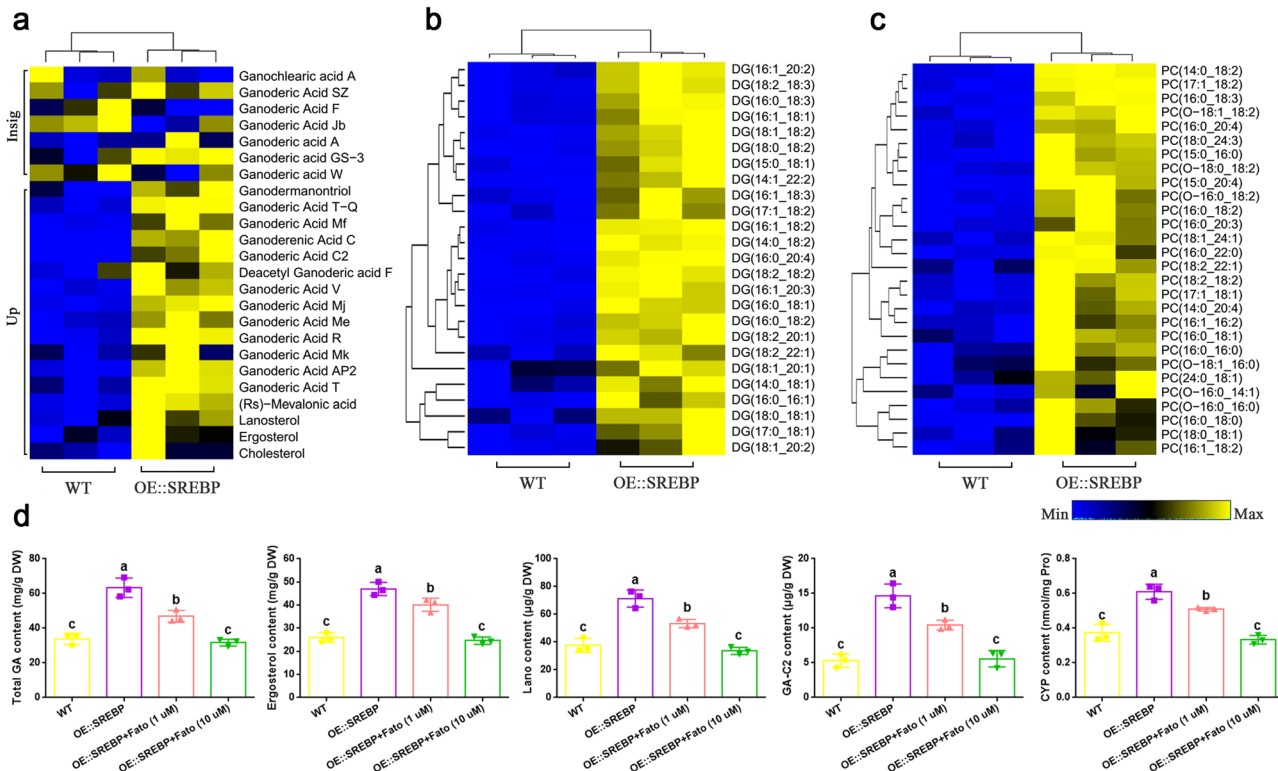

**Fig. 5 The roles of SREBP in metabolic abundance of GAs, sterols, and lipids. a** Heatmap showing the changes in the contents of detected GAs and sterols in WT and OE::SREBP strains. Up and Insig were indicated the significantly upregulated and no significantly differential metabolites, respectively, in the OE::SREBP strain compared to WT strain. Representative glycerolipids and glycerophospholipids, DG and PC, are shown in **b** and **c**, respectively. The significant differential metabolites had variable importance in projection (VIP) ≥ 1 and fold change ≥1.5 or ≤0.67. **d** The contents of cellular total GA, ergosterol, lanosterol (abbreviated as lano), GA-C2 and CYP in WT and OE::SREBP strains. 1 μM and 10μ M fatostatin were added at shaking for 5 days and then maintained until 7th days in liquid cultures of OE::SREBP strains. The values are the means ± SD of three independent experiments. Different letters indicate significant differences between the lines (n = 3 independent experiments, P < 0.05, according to Duncan's multiple range test).

CYPs (_g4875_ and _g4164_) were significantly upregulated in the OE::SREBP strain compared to the WT strain (Fig. 4d).

To validate our transcriptomics data, 10 up-DEGs belonging to the SREBP targets mentioned above were selected to determine the mRNA expression levels by qRT–PCR. The results show that the gene expression levels of 10 selected SREBP targets were significantly upregulated, ranging from 3.9- to 22.6-fold in the OE::SREBP strain (Fig. 4e), which is consistent with the RNA-seq results. Fatostatin, a specific inhibitor of SREBP, can impair the activation of SREBP[35,36]. To further validate the up-regulation role of SREBP in the 10 SREBP targets, fatostatin was added to the fermentation medium of OE::SREBP strain for detecting mRNA expression levels. As illustrated in Fig. 4e, an significantly decrease in gene expression levels of ten selected SREBP targets were observed when the exogenous fatostatin was added in OE::SREBP strain. A 10 μg/mL fatostatin treatment in OE::SREBP strain led to not significantly different in mRNA expression levels

than that of WT strain. These results indicate that SREBP positively regulates the expression of triterpenoid, ergosterol and lipid biosynthesis genes.

To further elucidate the role of SREBP in regulating triterpenoid, ergosterol and lipid biosynthesis, we performed targeted secondary metabolic profiling and lipid metabolic profiling between the WT and OE::SREBP strains. We identified 35 differential secondary metabolites including 29 upregulated, and 251 differential lipid metabolites, including 244 upregulated (VIP ≥ 1, and fold change ≥1.5 or ≤0.67), in the OE::SREBP strain compared to the WT strain (Supplementary Data 9). In particular, we observed that the contents of mevalonic acid, lanosterol, and 13 different GAs were significantly upregulated in the OE::SREBP strain (Fig. 5a). Significant accumulation of 25 diacylglycerol (DG) species and 28 phosphatidylcholine (PC) species was observed in the OE::SREBP strain (Fig. 5b and c).

Furthermore, the cellular total GA, ergosterol, lanosterol, and GA-C2 contents were determined in WT and OE::SREBP strains. As shown in Fig. 5d, the total GA, ergosterol, lanosterol, and GA-C2 contents of the OE::SREBP strain were significantly increased by ~1.87-, 1.84-, 1.89-, and 2.75- fold, respectively, compared to the WT, which is consistent with the metabonomics data results. In addition, the effects of overexpression SREBP on GA, ergosterol, lanosterol, and GA-C2 contents were recovered when OE::SREBP strain were treated with the SREBP inhibitor fatostatin. A 10 µg/mL fatostatin treatment in OE::SREBP strain led to not significantly different in total GA, ergosterol, lanosterol, and GA-C2 contents than that of WT strain (Fig. 5d). These above metabolic results is consistent with the effects of overexpression SREBP on gene expression. Moreover, we analyzed cellular CYP content to confirm the role of SREBP on GA biosynthesis. Overexpression SREBP led to a significantly increase of ~1.63-fold in the CYP content compared with the WT strain. The effects of overexpression SREBP on CYP content were also recovered when OE::SREBP strain were treated with the 10 µg/mL fatostatin (Fig. 5d).

Taken together, these results showed that overexpression of SREBP led to significant increases in GA (triterpenoid), ergosterol and lipid metabolism by facilitating transcription of target genes related to the GA, ergosterol and lipid biosynthesis.

## Discussion
The biosynthetic regulation mechanism of secondary metabolism has received widespread attention. As important pharmacologically active compounds of *Ganoderma* spp., GAs are also a hot research topic regarding its biosynthesis and related regulation mechanism, but the transcription factors directly regulating the expression of GA biosynthesis genes are still unknown. Here, genomics, DAP-seq, RNA-seq and metabolome techniques, as well as molecular biology and genetic methods, were integrated to clarify the molecular mechanism of SREBP in enhancing GA, ergosterol and lipid biosynthesis in the medicinal basidiomycete *G. lingzhi*. To the best of our knowledge, this is the first report to reveal a newly unknown function of SREBP in regulating GA biosynthesis in the fungus *G. lingzhi* and to demonstrate that SREBP enhances GA, ergosterol and lipid metabolism by binding promoter regions and facilitating the transcription of target genes related to the GA, ergosterol and lipid biosynthesis.

13 different GAs were significantly upregulated in the OE::SREBP strain, including Ganoderic acid C2, T-Q, T and ganodermanontriol. Ganoderic acid C2 has been used as an internal standard compound for the quality control of *Ganoderma* species[37]. Previous studies have indicated that ganoderic acid C2 possesses potential antihistamine, antitumour, anti-ageing and cytotoxic properties[38]. Ganoderic acid C2 also had the highest α-glucosidase inhibitory activity among the 12 tested ganoderic acids[39]. Ganoderic acid T significantly inhibits tumour invasion in vitro and in vivo through inhibition of matrix metalloproteinase expression[40]. Ganoderic acid T-Q, and ganodermanontriol were found to have the highest microtuble-stabilizing activity which is an important target for anticancer therapies[41]. In addition, ganodermanontriol also were found to be active as anti-HIV-1 agents[42]. Our results presented a potential strategy to improve GA production and *Ganoderma* pharmacological through the genetic manipulation of the key target SREBP in *G. lingzhi* and related *Ganoderma* species.

Extensive studies of the SREBP pathway in a variety of organisms have revealed that in addition to regulating lipid and sterol homoeostasis in animals, SREBP plays additional roles in the regulation of environmental adaptation. In fission yeast *Saccharomyces pombe*, SREBP pathway responds to sterols and functions as an oxygen sensor[43]. The lack of oxygen in *S. pombe* induces a drop in cholesterol synthesis which in turn activates SREBP-mediated transcription, and SREBP were able to upregulate sterol synthesis by targeting the activity of HMG-CoA reductase upon heat shock[44]. In *N. crassa*, the function of the SREBP pathway in ergosterol biosynthesis and adaptation to hypoxia was conserved, and activation of the SREBP pathway under lignocellulolytic conditions repressed a set of genes predicted to be involved in the endoplasmic reticulum stress response[45]. Previous studies have shown that heat stress leads to significant changes in phospholipid homoeostasis and increased GA content in *Ganoderma* species[19]. How SREBP regulates the synthesis of GAs, sterols, and lipids by taking part in the adaptation of *G. lingzhi* to heat stress is very interesting, and it is worth further study.

The SREBP-bound DNA motif has been described in model organisms previously. The sequence 5′-TCACNCCAC-5′ has been identified in humans as the SREBP binding motif[46,47]. A similar DNA motif, 5′-(A/G)TCA(T/C/G)(C/G)CCAC(T/C)−3′, was discovered as an SREBP DNA binding motif in the fungus *Aspergillus fumigatus* but was not similar to our sequence[48]. In fission yeast *S. pombe*, an SREBP-bound motif was defined using MEME to be (A/G)(C/T)C(A/G/T)NN(C/T)(C/T/G)A(C/T)[49], and a bHLH transcription factor (BHLH80) was found to bind the sequence TGCAAGT(G/T)G(A/C)(A/C/T) in *Arabidopsis thaliana*[50], which contains conserved residues similar to our sequence. Gel mobility shift experiments suggested that SREBP can bind a DNA sequence containing GAGGAG at the 3' end in HepG2 cells[51], which was similar to the best-fit motif of SREBP in *G. lingzhi*. Additionally, the bHLH transcription factors in *Homo sapiens* (NHLH) and *Drosophila melanogaster* (HLH4C) were found to bind 18-bp (5′-NDGNMKCAGCTGCGYCMH-3′) and 15-bp (5′-RGSMYCAGCTGCGYY-3′) binding regions, respectively, beginning with GGG, which contains similar initial conserved residues as our best-fit motif[52,53]. Overall, we reported a previously undescribed SREBP-bound motif, 5′-GRVGRVGRVGR-3′, in *G. lingzhi*, which provides a reference for studying the target gene binding the DNA sequence of the SREBP pathway.

Animal cells coordinate lipid homoeostasis by end-product feedback regulation of transcription through the proteolytic release of transcriptionally active SREBPs from intracellular membranes. The genes encoding sterol synthesis-related enzymes, such as 3-hydroxy-3-methylglutaryl CoA synthetase, 3-hydroxy-3-methylglutaryl CoA reductase, mevalonate kinase, squalene synthase, lanosterol synthase, and lanosterol 14α-demethylase, are targets for SREBPs in mammalian cells but not in *Drosophila*[29,54]. In the fungus *A. fumigatus*, SREBP also directly regulates ergosterol biosynthesis by 14-alpha sterol demethylase and C-4 methyl sterol oxidase[48]. In this study, 3-hydroxy-3-methylglutaryl CoA synthetase (*g1941*), mevalonate kinase (*g3941*), 14-demethylase (*g3908*), and sterol isomerase (*g7787*) were identified as SREBP target genes in *G. lingzhi*. Overexpression of SREBP led to significant increases in the contents of mevalonic acid, lanosterol, ergosterol and GAs and promoted the gene expression levels of the above SREBP targets. These results suggest that SREBP is conservative when regulating sterol synthesis in mammals and fungi but not in *Drosophila*. Similar studies have also been carried out in filamentous fungus *Aspergillus terreus*, in which knockout of the SREBP led to depression in sterol biosynthesis, but interestingly the raw materials of polyketides, such as acetyl-CoA and malonyl-CoA, became more available to increases production of polyketide lovastatin[55]. In addition, SREBP both promotes the synthesis of carotenoids and sterols which are synthesized by the MVA pathway in basidiomycete yeast *Xanthophyllomyces dendrorhous*[31]. A possible hypothesis is that SREBP can promote

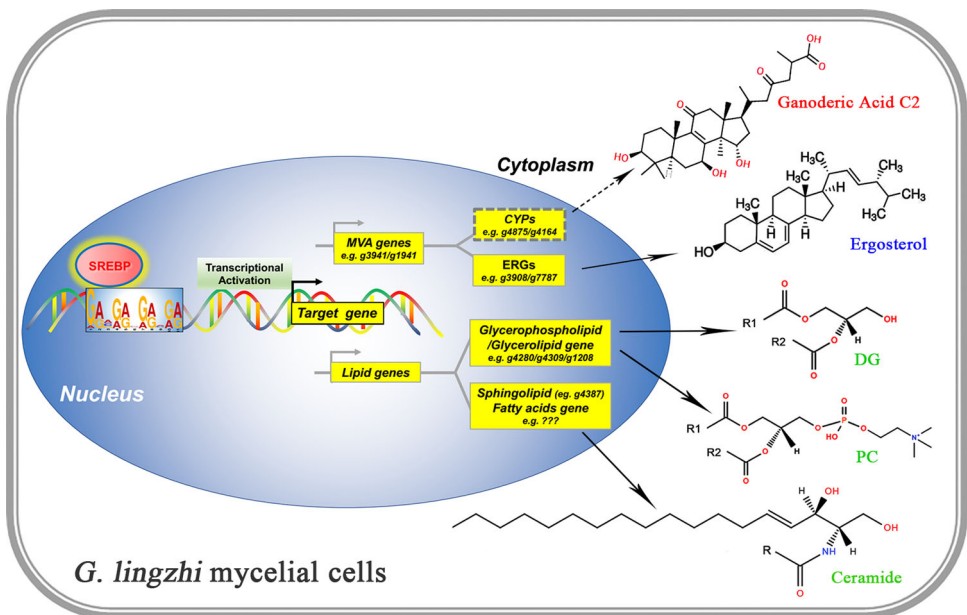

**Fig. 6 Model depicting the mechanisms of SREBP in enhancing GAs, ergosterol and lipid biosynthesis in *G. lingzhi*.** The bHLH-zip transcription factor SREBP interacts directly with the promoter binding sites (the best-fit motif 5′-GRVGRVGRVGR-3′ is shown) of SREBP targes. SREBP promoted the transcription of its target genes that are involved in GAs, ergosterol and lipid biosynthesis. The key target genes *g3941/g1941* encoding mevalonate kinase/ 3-hydroxy-3-methylglutaryl CoA synthetase, *g4875/g4164* encoding CYPs, *g7787/g3908* encoding sterol isomerase/lanosterol 14-demethylase, *g4280/g4309/g1208* encoding cardiolipin-specific phospholipase/ lysophospholipid acyltransferase/ethanolamine-phosphotransferase, and *g4387* encoding ceramide synthase, involved in MVA pathway, GAs biosynthesis, ergosterol biosynthesis, glycerolipid/glycerophospholipid biosynthesis, and sphingolipids biosynthesis, respectively, are displayed in the corresponding position. Two uncharacterized SREBP target CYPs, *g4875* and *g4164*, which possible involved in GA biosynthesis, marked by dotted line. The representative GAs (GA-C2), ergosterol, glycerolipid (DG), glycerophospholipid (PC), and sphingolipid (ceramide) are displayed. R, R1, and R2 indicate fatty acids.

a variety of secondary metabolites synthesized through MVA pathway.

In *Drosophila melanogaster*, a cholesterol auxotroph, SREBP is not regulated by sterols; instead, activation is controlled by phosphatidylethanolamine. In addition, other compounds, such as sphinganine, ceramide and hexadecanal, can also suppress SREBP activation cleavage when added to *Drosophila* cells in culture[30]. These results imply that *Drosophila* SREBP regulates the biosynthesis of glycerophospholipids and sphingolipids through unknown target genes. Our results revealed six genes involved in glycerophospholipid/glycerolipid biosynthesis, especially three genes (*g4280/g4309/g1208*) upregulated in OE::SREBP strains and a gene coding ceramide synthase (*g4387*) upregulated in OE::SREBP strains related to sphingolipid biosynthesis in *G. lingzhi* (Figs. 3 and 4). In addition, overexpression of SREBP led to significant increases in the contents of various glycerolipids, glycerophospholipids, and sphingolipids (Fig. 5). These results fill the unknown target gene of SREBP regulating glycerolipid, glycerophospholipid, and sphingolipid biosynthesis. On the other hand, the fatty acid synthesis genes acetyl-CoA synthetase, fatty acid synthase, and fatty acyl CoA synthetase are targets for SREBPs in mammals and *Drosophila*[30], but were not identified as SREBP targets in *G. lignzhi*. However, the overexpression of SREBP led to significant increases in the transcription level of fatty acid biosynthesis genes (Fig. 4c) and the contents of various triacylglycerols and diacylglycerols (Fig. 5), which is similar to the results of overproduction of triacylglycerol and fatty acids in transgenic mice expressing truncated SREBP[56].

Screening of cytochrome CYP gene candidates from *Ganoderma* spp., which may be responsible for GA biosynthesis from lanosterol but have not been functionally characterized. Seventy-eight CYP genes are coexpressed with lanosterol synthase in *G. lucidum*, and 16 of these genes show high similarity to fungal CYPs that specifically hydroxylate testosterone[22]. Combining *Vitreoscilla haemoglobin* (VHb) expression and calcium ion induction was developed to enhance GA production, and the transcription levels of three candidate CYPs, *cyp512a2*, *cyp512v2* and *cyp512a13*, were also increased under combined induction conditions in *G. lingzhi*[57]. Overexpression of the CYP gene *cyp5150l8* from *G. lucidum* was first found to produce the anti-tumor GA 3-hydroxy-lanosta-8,24-dien-26 oic acid (HLDOA) in *Saccharomyces cerevisiae*, as confirmed by HPLC, LC–MS and NMR[14]. In this study, 7 uncharacterized CYPs were identified as SREBP target genes, in which 2 CYPs, *g4875* and *g4164*, were coupregulated with 3-hydroxy-3-methylglutaryl CoA synthetase, mevalonate kinase and other MVA genes in the SREBP overexpression strain (Fig. 4d and e), strongly suggesting the possible roles of CYPs *g4875* and *g4164* in GA biosynthesis (Fig. 6, dashed box and arrow).

In summary, in this study, we identified a bHLH-zip transcription factor in the genome, SREBP, associated with GA, ergosterol, and lipid biosynthesis in the medicinal fungus *G. lingzhi* (Fig. 6). The DNA sequence 5′-GRVGRVGRVGR-3′ was identified as the best SREBP-binding motif, and the key target genes involved in GA, ergosterol, and lipid biosynthesis were revealed. Further genetic tests in which SREBP overexpression strains were constructed demonstrated that SREBP enhanced GA, ergosterol and lipid metabolism by binding the promoter regions and facilitating the transcription of target genes related to the GA, ergosterol and lipid biosynthesis.

## Methods

**Fungal strains and culture conditions**. Ganoderma strain SCIM 1006 (NO. CGMCC 18819) was selected for genome sequencing and was assigned to *G. lingzhi*[26]. The culture was maintained on artificial medium with shaking at 160 rpm in the dark. The culture medium was composed of the following (in g/L): glucose

(44.0), corn flour (0.5), peptone (6.5), $KH_2PO_4$ (0.75), $MgSO_4 \cdot 7H_2O$ (0.45), and vitamin $B_1$ (0.01)[16].

**Genome sequencing, assembly, and annotation**. A DNA library with 350-bp inserts was constructed and sequenced under an Illumina HiSeqX-Ten. For the PacBio RSII platform, a 20-kb library was generated and sequenced. The genome size of *G. lingzhi* was estimated by the k-mer method using sequencing data from the Illumina DNA library. Quality-filtered reads were subjected to 17-mer frequency distribution analysis using Jellyfish v2.2.10[58]. Then, the genome size, heterozygosity and repeat content were estimated by Genome-Scope web tools[59]. The de novo assembly of contigs was performed with FALCON (version 0.7.0). Pilon (v 1.23) was utilized to further correct the PacBio-corrected contigs with accurate Illumina short reads and to generate the genome assembly of *G. lingzhi*. The genome assembly was evaluated using BUSCO 3.1.0 (Benchmarking Universal Single-Copy Orthologues) with comparison to the lineage dataset fungi_odb9 (Creation date: 2016-10-21, number of species: 85, number of BUSCOs: 290).

For genome annotation, both homologous comparison and de novo prediction were used to annotate the repeated sequences within the *G. lingzhi* genome. A more detailed description can be found in previous reports[60]. The assembly reported in this paper is associated with NCBI BioProject: PRJNA738334 and BioSample: SAMN19718602. The data have been deposited in NCBI's Sequence Read Archive under accession numbers SRR14933280 and SRR14933281.

**DNA affinity purification sequencing (DAP-seq)**. *DAP-seq genomic DNA library preparation*: Fresh *G. lingzhi* mycelia were ground to fine powder using liquid nitrogen. DNA was extracted by CTAB and dissolved in Tris-EDTA buffer. DNA libraries were constructed by fragmenting genomic DNA to an average of 200 bp using a Covaris M220 (Woburn, MA, USA) according to the manufacturer's recommended setting. *DAP-seq protein expression*: The coding sequence of the SREBP bHLH domain was cloned into a pFN19K HaloTag T7 SP6 Flexi expression vector[61]. The halo–SREBP bHLH fusion protein was expressed using the TNT SP6 Coupled Wheat Germ Extract System (Promega) following the manufacturer's specifications and was directly captured using Magne Halo Tag Beads (Promega). *DAP-seq binding assay and sequencing*: The protein-bound beads were incubated with gDNA fragments, and the bound DNA fragments were released by heating to 98 °C. The bound DNA fragments were amplified by employing the KAPA HiFi HotStart ReadyMixPCR Kit (Roche, Basel, Switzerland). The PCR product was purified using AMPure XP beads (Beckman) and sequenced on an Illumina NavoSeq. *DAP-seq data processing*: Reads were mapped to the *G. lingzhi* genome sequence (this study) using BOWTIE2[62]. The reads are enriched at a certain location, which is called the peak. Peak calling was conducted using Macs2[63]. Associations of DAP-seq peaks located upstream or downstream of the transcription start site within 3.5 kb were analysed using Homer[64], based on the general feature format (gff) files. Gene function annotation was blasted against the NR, NT, SwissProt, and Pfam databases. FASTA sequences were obtained using BEDTools for motif analysis[65]. Motif discovery was performed using MEME-Chip suite 5.0.5[34].

**Electrophoretic mobility gel shift assay (EMSA)**. The coding sequence of SREBP-bHLH was cloned and recombined into the pTriEx4 vector (His-tag). The construct was transferred to *Escherichia coli* (BL21) for recombinant protein production. In parallel, nucleotide sequences were biotin labelled at the 3′ end using an EMSA Probe Biotin Labelling Kit (Beyotime, Nantong, China). Unlabelled probes were subjected to cold competition experiments. The nucleotide sequences of all probes used are shown in Supplementary Data 10. EMSA was performed using the Lightshift Chemiluminescent EMSA Kit (Thermo Scientific, 20148) according to the manufacturer's instructions. The biotin signals were imaged using the ChemiDoc MP Imaging System (Bio–Rad Laboratories, Inc., Hercules, CA, USA). The EMSAs were repeated three times, and representative results are shown.

**Construction of SREBP-overexpression plasmids and strains**. The SREBP overexpression vector (GLgpq-SREBP) was derived from the *Agrobacterium tumefaciens* binary vector pCAMBIA 1300 (CAMBIA, Canberra), which drives the expression of the hygromycin resistance gene (HygR) by 35 S promoter from CaMV and glyceraldehyde-3-phosphate dehydrogenase (GPD) gene promoter from *G. lingzhi*, respectively (Supplementary Fig. 4a). This vector was transformed to the *G. lingzhi* strain by *Agrobacterium tumefaciens*-mediated transformation (ATMT)[66], and the transformants were named OE::SREBP. The SREBP gene was amplified with the primers listed in Supplementary Data 10. A fusion fragment containing the Gl-gpd promoter and the SREBP gene were detected using the primers listed in Supplementary Data 10. qRT–PCR was performed to detect the expression of the SREBP gene in the WT strain and positive transformants (Supplementary Fig. 4).

**RNA-seq**. The WT and OE::SREBP strains were cultured in medium at 27 °C for 7 d with shaking at 160 rpm in dark. The total RNA of each sample was extracted from the fresh fermented mycelium of *G. lignzhi* according to the instruction manual of TRIzol Reagent (Life Technologies, California, USA). mRNA was isolated by the NEBNext Poly (A) mRNA Magnetic Isolation Module (NEB, E7490).

The cDNA library was constructed following the manufacturer's instructions of the NEBNext Ultra RNA Library Prep Kit for Illumina (NEB, E7530) and NEBNext Multiplex Oligos for Illumina (NEB, E7500).

Transcriptome analysis using reference genome-based read mapping: low-quality reads, such as those with adaptors only, unknown nucleotides >5%, or a Q20 < 20% (the percentage of sequences with sequencing error rates <1%), were removed by Perl script. The clean reads filtered from the raw reads were mapped to the *G. lingzhi* genome (this study) using Tophat2 software[67]. The aligned records from the aligners in BAM/SAM format were further examined to remove potential duplicate molecules. Gene expression levels were estimated using FPKM values (fragments per kilobase of exon per million fragments mapped) by Cufflinks software[68].

Identification of differential gene expression: differential expression analysis between WT and OE::SREBP strains was performed using the DESeq2[69]. DESeq2 based on raw counts had been used for obtaining DEG and the raw counts value statistics using StringTie[70]. The resulting *P*-values were adjusted using the Benjamini and Hochberg's approach for controlling the false discovery rate (FDR). The FDR < 0.01 and fold change ≥ 2 was set as the threshold for significantly differential expression.

**Real-time quantitative PCR analysis of gene expression**. The levels of gene-specific mRNA expressed were assessed using qRT-PCR, according to the previous study[71]. Briefly, total RNAs were separately extracted and reversed into cDNAs. Gene expression was evaluated by calculating the difference between the threshold cycle (CT) value of the gene analysed and the CT value of the housekeeping gene *18 S* rRNA. qRT-PCR calculations analysing the relative gene expression levels were performed according to the $2^{-\Delta\Delta CT}$ method with paired primes listed in Supplementary Data 10.

**Western blotting**. Preparation of anti-SREBP antibody and western blotting was performed as previously described[71]. Briefly, polyclonal antibody against SREBP was obtained by sent the SREBP-bHLH protein (same as the protein used for EMSA analysis above) to a professionally qualified antibody preparation company and used for the immunization of rabbits (Chemgen Biotech, Shanghai, China). Proteins from WT and OE::SREBP strains of mycelia were separated in a 12% (w/v) SDS-PAGE gel, transferred to polyvinylidene difluoride membranes (Bio-Rad), and incubated with a primary rabbit anti-SREBP antibody to detect SREBP and then with a secondary HRP goat anti-rabbit IgG antibody. β-Actin was used as internal reference and detected with a primary mouse β-Actin-specific antibody (1:2000, AT0097, CMCTAG) and a secondary HRP goat anti-mouse IgG antibody.

To separate the total mycelium extract into cytosolic and nuclear protein fractions, the NE-PER Nuclear and Cytoplasmic Extraction Reagents were used (Thermo Fisher Scientific, IL, USA). β-Tubulin antibody (1:2000, AT0003, CMCTAG) and Histone-H3 antibody (1:2000, AT0005, CMCTAG) were used as cytoplasmic and nuclear internal reference, respectively.

**Secondary metabolite extraction and mass spectrometry analysis**. Biological samples were freeze-dried by a vacuum freeze-dryer (Scientz-100F). The freeze-dried sample was crushed using a mixer mill (MM 400, Retsch) with a zirconia bead for 1.5 min at 30 Hz. Then, 100 mg of lyophilized powder was dissolved in 1.2 mL 70% methanol solution, vortexed for 30 s every 30 min for a total of 6 times, and finally placed in a refrigerator at 4 °C overnight. Following centrifugation at 12000 rpm for 10 min, the extracts were filtered (SCAA-104, 0.22-μm pore size; ANPEL, Shanghai, China, http://www.anpel.com.cn/) before UPLC–MS/MS analysis.

The sample extracts were analysed using an LC–ESI–MS/MS system (UPLC, Shim-pack UFLC SHIMADZU CBM A system, https://www.shimadzu.com/; MS, QTRAP® 4500+ System, https://sciex.com/). The analytical conditions were as follows: UPLC: column, Waters ACQUITY UPLC HSS T3 C18 (1.8 μm, 2.1 mm*100 mm); column temperature, 40 °C; flow rate, 0.4 mL/min; injection volume, 2 μL; solvent system, water (0.1% formic acid): acetonitrile (0.1% formic acid); gradient programme, 95:5 V/V at 0 min, 5:95 V/V at 10.0 min, 5:95 V/V at 11.0 min, 95:5 V/V at 11.1 min, and 95:5 V/V at 15.0 min.

LIT and triple quadrupole (QQQ) scans were acquired on a triple quadrupole-linear ion trap mass spectrometer (Q TRAP), the AB4500 Q TRAP UPLC/MS/MS System, equipped with an ESI Turbo Ion-Spray interface operating in positive and negative ion mode and controlled by Analyst 1.6.3 software (AB Sciex). The ESI source operation parameters were as follows: ion source, turbo spray; source temperature 550 °C; ion spray voltage (IS) 5500 V (positive ion mode)/-4500 V (negative ion mode); ion source gas I (GSI), gas II (GSII), and curtain gas (CUR) were set at 50, 60, and 25.0 psi, respectively; and collision-activated dissociation (CAD) was high. Instrument tuning and mass calibration were performed with 10 and 100 μmol/L polypropylene glycol solutions in QQQ and LIT modes, respectively. QQQ scans were acquired as MRM experiments with collision gas (nitrogen) set to medium. DP and CE for individual MRM transitions were performed with further DP and CE optimization. A specific set of MRM transitions was monitored for each period according to the metabolites eluted within this period. MS/MS data of identified GAs was shown in Supplementary Data 11.

**Lipid extraction and mass spectrometry analysis**. The freeze-dried mycelium was crushed using a mixer mill (Tissuelyser-48, http://www.tissuelyser.com/) with a zirconia bead for 1.5 min at 50 Hz. Fifty (±1) milligrams of powder was weighed and supersonically extracted with 1.0 mL isopropanol (IPA) for 10 min. Following centrifugation at 12,000 rpm for 10 min, the extracts were concentrated and then dissolved in 200 µl IPA before LC–MS analysis.

The sample extracts were analysed using an LC–ESI–MS/MS system. For separation, a Thermo Accucore™ C30 column (2.6 µm, 2.1 × 100 mm) was used. Mobile phase A consisted of water/acetonitrile (V/V = 4:6) with 10 mM ammonium formate and 0.04% acetic acid, mobile phase B (consisted of isopropanol/acetonitrile (V/V = 9:1) with 10 mM ammonium formate and 0.04% acetic acid) with the flow rate set at 0.35 mL/min, and the column temperature maintained at 40 °C. Separation was achieved by using a gradient starting at 20% B, which was increased to 30% B at 2 min, increased to 65% B at 4 min and held for 4 min, increased to 85% B at 9 min, increased to 90% B at 14 min, increased to 95% B at 15.5 min and held for 2 min, followed by a change back to 20% B at 0.2 min with holding for 2.3 min. The injection volume for samples was 3 µL. The same mass spectrometry method as the above method used in sterol was used for lipid analysis.

**Determination of cellular total GA, ergosterol, lanosterol, GA-C2 and CYP contents**. For the determination of total GA content, the dried mycelia (2 g) were extracted by circumfluence with 75% (v/v) ethanol (100 mL) for 3 h (twice). After removal of the mycelia by centrifugation, the supernatant was dried under vacuum. The residues were suspended in water and later extracted with chloroform (100 mL) for 2 h (twice). After removal of the chloroform by evaporation, the sample was further extracted with 5% (w/v) NaHCO₃ (200 mL) for 12 h, and adding 2 M HCl to adjust the pH to 3. The GAs in the NaHCO₃ layer were extracted with chloroform (200 mL) for 12 h. After removal of the chloroform by evaporation, the GAs were then dissolved in absolute ethanol, and their absorbance was measured at 245 nm using ursolic acid as the standard.

The CYP content was determined using reduced carbon monoxide (CO) difference spectroscopy[16]. The cell lysate was solubilized in 0.1 M phosphate buffer (pH 7.4) with 20% glycerol and was distributed equally into two quartz cuvettes. 10% solid sodium dithionite was added into each cuvettes and mixed well. The baseline absorbance was recorded using a double beam ultravioletvisible (UV-Vis) spectrophotometer from 400 to 500 nm wavelength. The concentration of CYP ($c$) in the cuvette was calculated using the Beer's law equation: $c = A/(\varepsilon \times L)$, where $A$ is the difference in light absorbance between 450 and 490 nm, $L$ is the light path of the cuvette (1 cm), and $\varepsilon$ is the extinction coefficient ($91\ \mathrm{mM^{-1}\ cm^{-1}}$).

For the determination of ergosterol and lanosterol, the sterols were extracted with methanol and ethanol (60:40, v/v) (three times). The extracts were saponified with 0.1 M methanolic NaOH at 50 °C for 2 h. The hydrolysed samples were mixed with 2 mL of distilled deionized water and extracted twice with 5 mL petroleum ether (boiling point range, 60–90 °C). The petroleum ether layer was pooled and evaporated to dryness under a stream of nitrogen. The dry samples were redissolved in 100 µL of methanol and were later injected into an Agilent 1200 series HPLC with an Agilent Zorbax SB-C18 column (250 × 4.6 mm, 5 µm). The detector was set at 210 and 282 nm. Chromatographic peaks were identified by comparing the retention times and spectra against the standards of lanosterol (≥98%, MedChemExpres) and ergosterol (>98%, MedChemExpres)[9].

For the measurement of individual ganoderic acids (GA-C2), 100 mg dried mycelia was extracted with methanol, and the GA-C2 in the supernatant were monitored at 254 nm by HPLC using an Agilent 1200 series HPLC with an Agilent Zorbax SB-C18 column (250 × 4.6 mm, 5 µm). The calibration curve for the measurement of GA-C in the fungal mycelium was constructed using the standards of GA-C2 ( >99%, MedChemExpress)[72].

**Statistics and reproducibility**. All data presented in this manuscript are from three independent experiments. Error bars indicate standard deviation from the mean from triplicate independent experiments. Two methods were used to analyse the significance of the data. Asterisks indicate significant differences (**$P < 0.01$) compared to the control according to two-way analysis of variance (ANOVA) using GraphPad Prism. Different letters indicate significant differences between the lines ($P < 0.05$) according to Duncan's multiple range test.

**Reporting summary**. Further information on research design is available in the Nature Research Reporting Summary linked to this article.

## Data availability
The genome assembly reported in this paper is associated with NCBI BioProject: PRJNA738334 and BioSample: SAMN19718602. The detailed sequencing information have been deposited in NCBI's Sequence Read Archive under accession numbers SRR14933280 and SRR14933281. An annotated gff3 file was shown in Supplementary Data 12. DAP-seq and RNA seq data underlying the findings described in the manuscript are fully available without restriction from the Bioproject Sequence Read Archive: SRR21374162-64 and SRR20305604-09, respectively. The raw data for Figs. 4e, 5d, S4d,

S4e were shown in Supplementary Data 13. Uncropped western blots are in Supplementary Fig. 6.

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

## Acknowledgements

We thank Shanghai OE Biotech. Co., Ltd of China for providing the Genome analysis, Bluescape Hebei Biotech Co., Ltd of China for providing the DAP-seq analysis, Beijing BioMarker Technologies of China for providing the RNA-seq analysis, and Wuhan HealthCare Metabolic Biotechnology Corporation of China for providing the metabolic analysis. This work was supported by grants from National Natural Science Foundation of China (31900027, 31772374 and 32071673), the China Postdoctoral Science Foundation (2020M682601), the Science and Technology Innovation Program of Hunan Province (2020RC2059 and 2021RC4063), the Natural Science Foundation of Hunan Province (2020JJ5972 and 2021JJ31151), the Scientific Research Fund of Hunan Provincial Education Department, China (No. 18B167).

## Author contributions

Y.-N.L. designed the study. Y.-N.L., F.-Y.W., R.-Y.T, Y.-X.S, Z.-Q.X, and J.-Y.L carried out experiments and analyzed data. J.H, F.-F.X, B.-Y.L, and G.-Q.L provided supervisor oversight. Y.-N.L. and G.-Q.L wrote the manuscript. All authors gave input and approved the manuscript.

## Competing interests

The authors declare no competing interests.
