## [Peer Review File · Communications Biology]

Reviewers' comments:

Reviewer #1 (Remarks to the Author):

The authors characterized the function of SREBP transcription factor in the mushroom spp. *Ganoderma lingzhi* using genome annotation, DAP-seq, RNA-seq and lipid profiling. SREBP transcription factors are well known in their role in sterol biosynthesis. So the findings reported of SREBP in GAs, ergosterol and lipid in this species are consistent with established function of SREBP transcription factors in other fungi. The presentation is clear and logic. I just have a few suggestions:

1. Since the authors can genetically manipulate *Ganoderma lingzhi*, it would be better to include ChIP-DNA seq to verify targets bound by this transcription factor in vivo. This should be achievable given that tagged version of SREBP in other fungal species is functional.
2. SREBP is activated by protease cleavage and the cleaved version translocates to the nucleus to function as a transcription factor. It will strengthen the study if the authors could demonstrate that overexpression indeed leads to increased active version of SREBP.
3. Provide accession number for DAP-seq and RNA seq data (SRA files and annotation file for its genome if not available at NCBI).

Reviewer #2 (Remarks to the Author):

The manuscript "The regulatory and transcriptional landscape associated with triterpenoid and lipid metabolisms by the bHLH-zip transcription factor SREBP in the medicinal fungus *Ganoderma lingzhi*" identified and described the transcription factor SREBP from *G. lingzhi*, a fungus that produces ganoderic acids (GA) that have important pharmacological properties. The authors identified several genes regulated by SREBP (through DAP-seq, RNA-seq, EMSA, and RT-qPCR), with some involved in the triterpenoid and lipid metabolism. Consequently, the overexpression of SREBP increased the production of triterpenoids, ergosterol, lipids, and of several different GAs. Treatment of the SREBP overexpression strain with fatostatin (that inhibits SREBP activation) returned the production of these metabolites to wildtype levels. The manuscript is well written, and it is easy to follow despite the complexity of the topic. The experimental approach was interesting, and the findings are very exciting. I enjoyed reading this manuscript. I only have minor comments:

L106-108: I am not sure that SREBP is conserved in bacteria. The cited manuscript refers to regulated intramembrane proteolysis, RIP; which is conserved in bacteria, but (to the best of my knowledge) SREBP activation is an example of RIP in eukaryotes. Please check.

L266: "...induces a drop in cholesterol..." Should it be ergosterol (not cholesterol)? Please check.

References: some journal names are abbreviated, and some are not. Please check the journal indications.

Legend Figure 4: Please indicate what CK and Fato stand for in the figure legend.

Figure 5:

- Figure 5A: I do not understand the "Insig" and "Up" separation in the figure.
- Figure 5A: *G. lingzhi* produces cholesterol? Is this correct? (bottom of figure 5A, please check).
- Figure 5D: Lano in the third graph means lanosterol? (this abbreviation should be indicated in legend).
- Just a personal appreciation, the heat map confuses me, showing red meaning more and green meaning less (as red is generally related to off and green with on).

Detailed Response to Reviewers

Reviewers' Comments & Our Responses:

Reviewer #1 (Remarks to the Author):

The authors characterized the function of SREBP transcription factor in the mushroom spp. *Ganoderma lingzhi* using genome annotation, DAP-seq, RNA-seq and lipid profiling. SREBP transcription factors are well known in their role in sterol biosynthesis. So the findings reported of SREBP in GAs, ergosterol and lipid in this species are consistent with established function of SREBP transcription factors in other fungi. The presentation is clear and logic. I just have a few suggestions:

Response: Special thanks to the reviewer for the positive evaluation.

1. Since the authors can genetically manipulate *Ganoderma lingzhi*, it would be better to include ChIP-DNA seq to verify targets bound by this transcription factor *in vivo*. This should be achievable given that tagged version of SREBP in other fungal species is functional.

Response: Thank you very much for your suggestion. We agree with you and believe that it would be better to include ChIP-DNA seq (ChIP-seq) to verify SREBP targets *in vivo*. ChIP-seq is the current leading method for determining *in vivo* transcription factor binding sites (TFBS) (Kheradpour and Kellis, 2014; Stamatoyannopoulos et al., 2012). However, ChIP-seq experiments have been generally limited in scale as they are difficult to execute, dependent on antibody quality, and challenging for rare or lowly expressed proteins (Kidder et al., 2011). In addition, ChIP-seq is limited in its throughput by the need to create gene-specific antibodies or tagged transgenic lines, which can be technically challenging and expensive (Landt et al. 2012). **As a result, binding site information is available for relatively few TFs and substantial TFBS coverage is only available for humans and several model organisms** (O'Malley et al., 2016). For *Ganoderma lingzhi* (a macrofungus), it is still not a model organism, and there is no reports on the application of ChIP-seq in *G. lingzhi* (to the best of my knowledge). Therefore, up to now, it is still a difficult experimental technique for ChIP-seq of SREBP applied in *G. lingzhi* although we have tried many

times before.

Alternative ChIP-seq methods are needed for capturing genome-wide binding data for many organisms. DAP-seq method is fast, inexpensive, and more easily scaled than ChIP-seq (Bartlett et al., 2017). DAP-seq has been applied to the study of TFBSs in many species. For example, in human pathogen *Pseudomonas aeruginosa*, the DAP-seq was used to map the genome-wide binding sites of two-component systems response regulators (Trouillon et al., 2021). In filamentous fungi *Neurospora crassa*, using DAP-seq, the direct targets of TF involved in regulating genes encoding plant cell wall-degrading enzymes were identified (Wu et al, 2020). In perennial tree, *Populus euphratica*, DAP-seq was performed to screen direct target genes of the transcription factor, PeWRKY1 (Yao et al., 2019). Therefore, we used DAP-seq to study SREBP targets in *G. lingzhi*. And, in the discussion section, SREBP targets in different studies were discussed.

On the other hand, although we only used DAP-seq (not ChIP-seq, due to technical difficulties) to explore potential targets, we also have applied EMSA validation, and adopted the overexpressing SREBP, RNA-seq and metabolome to support our topic.

Thank you again for your suggestion. And, we regret that we are unable to meet the ChIP-seq experimental suggestion due to technical difficulties. Please understand and sympathize with our difficulties.

References:

- Bartlett A, et al. Mapping genome-wide transcription-factor binding sites using DAP-seq. *Nat Protoc* 12, 1659-1672 (2017).
- Kheradpour P, Kellis M. Systematic discovery and characterization of regulatory motifs in ENCODE TF binding experiments. *Nucleic Acids Res* 42:2976–2987 (2014).
- Kidder BL, et al. ChIP-Seq: technical considerations for obtaining high-quality data. *Nat Immunol* 12:918–922 (2011).
- Landt SG, et al. ChIP-seq guidelines and practices of the ENCODE and modENCODE consortia. *Genome Res* 22:1813–1831 (2012).
- O'Malley RC, et al. Cistrome and epicistrome features shape the regulatory DNA landscape. *Cell* 165, 1280-1292 (2016).
- Stamatoyannopoulos, et al. Mouse ENCODE Consortium. An encyclopedia of mouse DNA elements (Mouse ENCODE). *Genome Biol* 13:418 (2012).
- Trouillon J, et al. Determination of the two-component systems regulatory network reveals core and accessory regulations across *Pseudomonas aeruginosa* lineages. *Nucleic Acids Res* 49(20):11476-11490 (2021). **(DAP-Seq is used to study TFBSs in bacteria)**

Wu V, et al. The regulatory and transcriptional landscape associated with carbon utilization in a filamentous fungus. *P Natl Acad Sci USA* 117.11:201915611 (2020). **(DAP-Seq is used to study TFBSs in fungi)**

Yao J, et al. *Populus euphratica* WRKY1 binds the promoter of PeHA1 to enhance gene expression and salt tolerance. *J Exp Bot* 71(4):1527-1539 (2019). **(DAP-Seq is used to study TFBSs in plant)**

2. SREBP is activated by protease cleavage and the cleaved version translocates to the nucleus to function as a transcription factor. It will strengthen the study if the authors could demonstrate that overexpression indeed leads to increased active version of SREBP.

Response: Thanks for the Reviewer's suggestion. We have analyzed the protein levels of SREBP in WT and OE::SREBP strains by Western blot (new Figure S4), and the results demonstrated that overexpression indeed leads to increased active version of SREBP (Figure S4E and F). The related methods and results have been described in the revised manuscript on page 9, lines 175-181; pages 21-22, lines 436-449; pages 40-41, lines 773-779.

The polyclonal antibodies against SREBP had been prepared by immunization of rabbits with the SREBP-bHLH proteins by a professionally qualified antibody preparation company in July, which were mainly used to study the function of SREBP in responding to the environment induced GAs biosynthesis.

3. Provide accession number for DAP-seq and RNA seq data (SRA files and annotation file for its genome if not available at NCBI).

Response: According to the reviewer's suggestion, we deposited DAP-seq and RNA-seq data in NCBI's Sequence Read Archive under accession numbers SRR21374162-64 and SRR20305604-09, respectively. An annotated gff3 file were shown in Supplementary materials 12. These informations were added to a separate Data Availability Section (Please see pages 24-25, lines 499-505)

Reviewer #2 (Remarks to the Author):

The manuscript "The regulatory and transcriptional landscape associated with triterpenoid and lipid metabolisms by the bHLH-zip transcription factor SREBP in the medicinal fungus *Ganoderma lingzhi*" identified and described the transcription factor SREBP from *G. lingzhi*, a fungus that produces ganoderic acids (GA) that have important pharmacological properties. The authors identified several genes regulated by SREBP (through DAP-seq, RNA-seq, EMSA, and

RT-qPCR), with some involved in the triterpenoid and lipid metabolism. Consequently, the overexpression of SREBP increased the production of triterpenoids, ergosterol, lipids, and of several different GAs. Treatment of the SREBP overexpression strain with fatostatin (that inhibits SREBP activation) returned the production of these metabolites to wildtype levels. The manuscript is well written, and it is easy to follow despite the complexity of the topic. The experimental approach was interesting, and the findings are very exciting. I enjoyed reading this manuscript. I only have minor comments:

Response: Special thanks to the reviewer for the positive evaluation.

1. L106-108: I am not sure that SREBP is conserved in bacteria. The cited manuscript refers to regulated intramembrane proteolysis, RIP; which is conserved in bacteria, but (to the best of my knowledge) SREBP activation is an example of RIP in eukaryotes. Please check.

Response: Thanks for the reviewer's suggestion. We have checked the cited manuscript and reviewed other SREBP related papers, and found that SREBP is rarely studied in bacteria. Whether SREBP is conserved in bacteria is uncertain. Therefore, we revised the manuscript and the corresponding references in the new lines 106-108 and new Ref. No. 28, respectively.

2. L266: "...induces a drop in cholesterol..." Should it be ergosterol (not cholesterol)? Please check.

Response: Thanks very much. We have checked the original literature and found that it's "cholesterol" not "ergosterol".

Robichon C, Dugail I. De novo cholesterol synthesis at the crossroads of adaptive response to extracellular stress through SREBP. *Biochimie*. 2007 Feb;89(2):260-4. doi: 10.1016/j.biochi.2006.09.015. Epub 2006 Oct 12. PMID: 17059860.

3. References: some journal names are abbreviated, and some are not. Please check the journal indications.

Response: Thanks. According to the reviewer's suggestion and journal indications, we have revised the names of all the journals as abbreviations. Please see journal names marked in red in References section.

4. Legend Figure 4: Please indicate what CK and Fato stand for in the figure legend.

Response: Thanks very much for your suggestion. We have indicated what CK and Fato stand for in the figure legend (Please see page 34, lines 710-712).

5. Figure 5:

5.1- Figure 5A: I do not understand the “Insig” and “Up” separation in the figure.

Response: We are sorry for the confusion. Twenty different GAs were identified, in which 13 different GAs were significantly upregulated in the OE::SREBP strain, the other showed no significant difference. To make it easier to understand, the “Insig” and “Up” had been indicated in figure legend (Please see page 35, lines 718-720).

5.2- Figure 5A: *G. lingzhi* produces cholesterol? Is this correct? (bottom of figure 5A, please check).

Response: Special thanks to the reviewer. According to the reviewer's suggestion, we have checked the results and found that a smaller amount of cholesterol was identified compared to ergosterol (Supplementary Materials 9). Similar results were also found in fungi *Neurospora crassa*, where a small amount of cholesterol was detected (Qin, *et al.*, 2017).

Qin L, Wu VW, Glass NL. 2017. Deciphering the regulatory network between the SREBP pathway and protein secretion in *Neurospora crassa*. mBio 8:e00233-17 (**Results: Fig. 2B**).

5.3- Figure 5D: Lano in the third graph means lanosterol? (this abbreviation should be indicated in legend).

Response: Yes, lano in the third graph means lanosterol. According to the reviewer's suggestion, the abbreviation had been indicated in legend of Figure 5 (Please see page 35, line 723).

5.4- Just a personal appreciation, the heat map confuses me, showing red meaning more and green meaning less (as red is generally related to off and green with on).

Response: We are very sorry for the confusion. We have changed the heatmap color and used yellow/blue color contrast to facilitate recognition by colorblind readers (Please see new Figure 5 A-C).

REVIEWERS' COMMENTS:

Reviewer #1 (Remarks to the Author):

Thanks for addressing my concerns.

Reviewer #2 (Remarks to the Author):

The revised version of the manuscript entitled "The regulatory and transcriptional landscape associated with triterpenoid and lipid metabolisms by the bHLH-zip transcription factor SREBP in the medicinal fungus *Ganoderma lingzhi*" addressed my previous comments satisfactorily. In addition, the authors included western blot assays in a supplementary figure that complement previous experiments. I only have two minor observations:

L438: Please check wording "...antibodies against SREBP were obtained by sent the SREBP-bHLH proteins..."

L711: Please correct "...OE::SRBEP..."